# *Scaling Laws vs Model Architectures*:
# How Does Inductive Bias Influence Scaling?

**Yi Tay**[*†]   **Mostafa Dehghani**[*]   **Samira Abnar**[†]   **Hyung Won Chung**[†]
**William Fedus**[†]   **Jinfeng Rao**[†]   **Sharan Narang**[†]   **Vinh Q. Tran**
**Dani Yogatama**[†]   **Donald Metzler**

Google

dehghani@google.com

## Abstract

There have been a lot of interest in the scaling properties of Transformer models (Kaplan et al., 2020). However, not much has been done on the front of investigating the effect of scaling properties of different inductive biases and model architectures. Do model architectures scale differently? If so, how does inductive bias affect scaling behaviour? How does this influence upstream (pretraining) and downstream (transfer)? This paper conducts a systematic study of scaling behaviour of ten diverse model architectures such as Transformers, Switch Transformers, Universal Transformers, Dynamic convolutions, Performers, and recently proposed MLP-Mixers. Via extensive experiments, we show that (1) architecture is an indeed an important consideration when performing scaling and (2) the best performing model can fluctuate at different scales. We believe that the findings outlined in this work has significant implications to how model architectures are currently evaluated in the community.

## 1 Introduction

There have been a lot recent interest in the scaling properties of Transformer models (Kaplan et al., 2020; Hernandez et al., 2021; Bahri et al., 2021; Henighan et al., 2020; Tay et al., 2021b; Abnar et al., 2021). However, not much is understood about the scaling properties of different inductive biases imposed by model architectures. Improvements at a a specific scale (compute, size etc) are often assumed to transfer to different scales and compute regions (So et al., 2019; Choromanski et al., 2020; Lan et al., 2019; Dehghani et al., 2018) and new research is often presented in a point-wise fashion with respect to scale. In short, it is not uncommon for new methods to be presented with data points at very specific or limited compute regions (e.g., base size). We believe that understanding the

interaction between architecture and scaling laws is crucial as designing models that perform well at diverse scales will likely have significant impact.

This paper is an attempt to understand the effect of inductive bias (architecture) on scaling laws of language models. To this end, we pre-train and finetune over ten diverse model architectures across multiple compute region and scales (e.g., from 15M to 40 Billion parameters). In total, we pre-train and finetune over 100 different models of different architectures and sizes and present insights and challenges at scaling these ten diverse architectures.

We consider a broad spectrum of models in our extensive experiments. Concretely, we consider several well-established Transformer variants (Vaswani et al., 2017) such as Evolved Transformer (So et al., 2019), Universal Transformers (Dehghani et al., 2018) and Switch Transformers (Fedus et al., 2021). We also consider lightweight models such as ALBERT (Lan et al., 2019) and/or efficient Transformers (Tay et al., 2020) such as Performer (Choromanski et al., 2020) and Funnel Transformers (Dai et al., 2020). In our comparison, we are also interested in finding out if general improvements to the Transformer architectures such as Mixture-of-Softmax (Yang et al., 2017) and/or Gated Linear Units (Dauphin et al., 2017; Shazeer, 2020) influence the scaling behaviour of models. Finally, we also evaluate models outside the family of Transformers including Lightweight convolutions (Wu et al., 2019), Dynamic convolutions (Wu et al., 2019) and the recently proposed MLP-Mixers (Tolstikhin et al., 2021). Figure 1 illustrates an overview about the experiments we run.

We also note that scaling these models is not as straightforward as it seems, i.e., there are intricate details of scale that are intertwined with architectural choices which we study in detail in this paper. For example, a distinct feature of Universal Transformers (and ALBERT) is parameter sharing. Hence, compared with standard Transformers, this

---
[*]Yi and Mostafa contributed equally.
[†]Work completed while at Google.

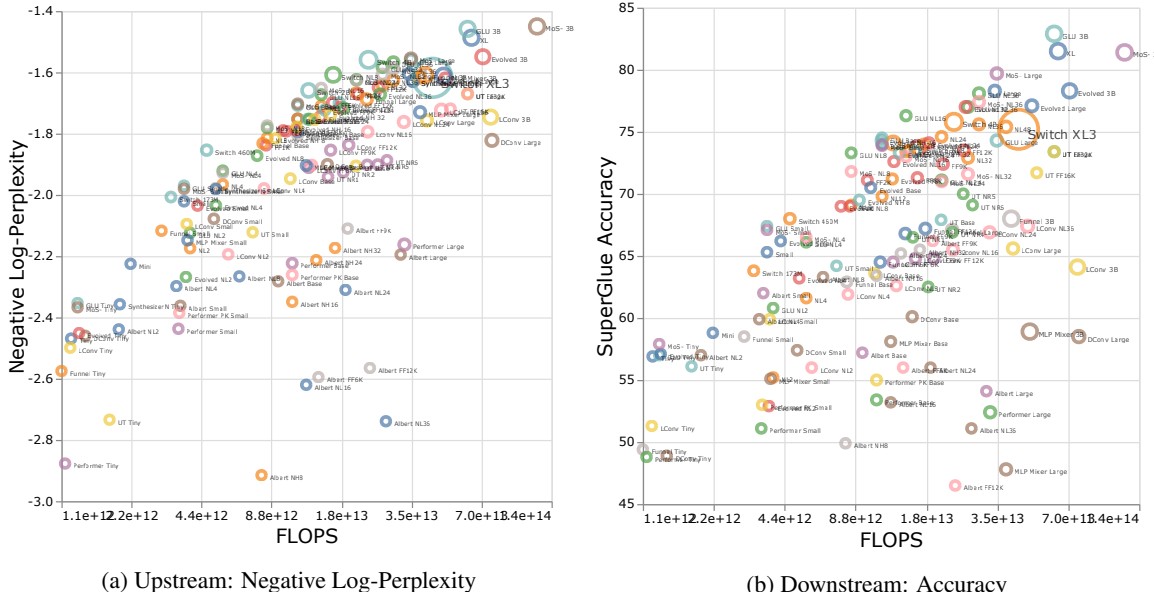

(a) Upstream: Negative Log-Perplexity

(b) Downstream: Accuracy

Figure 1: An overview compute-performance (FLOPs vs performance) plot of all the diverse models and architectures we pretrained and finetuned in this study. Colors represent different model architectures and size of the circles represent the size of the model (parameters).

architectural choice significantly warps the scaling behaviour not only with respect to performance but also amongst compute metrics such as FLOPs, speed and number of parameters (Dehghani et al., 2021a). Conversely, models such as Switch Transformers are on the other end of the spectrum with an uncommon relationship between FLOPs and number of parameters, i.e., they have high parameter to FLOPs ratio. This difficulty makes navigating this landscape challenging.

**Our Contributions and Insights** The key contributions of this paper are as follows:

- For the first time, we derive scaling laws for different inductive biases and model architectures. We find that this scaling coefficient differs greatly from model to model. We believe this is an important consideration in model development. It turns out that amongst all ten architectures that we consider, the vanilla Transformer has the best scaling behaviour, even if its absolute performance at each compute region is not the greatest.

- We observe that models that operate well in one compute-scale region is not necessarily the best in another compute-region. Moreover, we find that certain models have difficulty scaling despite performing decently (comparably) at lower-compute regions. This has implications, since it is difficult to get the fulll picture of a model's

scalability with pointwise comparisons at a certain compute-region.

- We find that when it comes to scaling different model architectures, upstream pre-training perplexity might not correlate well with downstream transfer. Hence, the underlying architecture and inductive bias is also crucial for downstream transfer.

- We highlight the difficulties of scaling with certain architectures and show that some models do not scale (or scale with a negative trend). We also find concerning trends where linear-time attention models such as Performer struggle with scaling up.

## 2 Related Work

Kaplan et al. (2020) studied empirical scaling laws of the decoder-only Transformer language models. They focused on the standard left-to-right language modeling objective with the cross-entropy loss as the performance metric. One of the main findings is that the loss scales as a power-law with three major characteristics of the model training: model size, dataset size and the training compute. Another somewhat surprising finding is that the model shapes such as width or depth of the Transformer network have minimal effects on the cross-entropy loss for a wide range of scales. Subsequent works (Henighan et al., 2020; Hernandez et al.,

2021) made similar conclusions for autoregressive generative modeling and for transfer learning, respectively. This finding is also generally supported by (Tay et al., 2021b) but discrepancies were found for the gap between pretraining and finetuning - highlighting the fact that observing downstream performance of large language model is indeed important. In (Tay et al., 2021b), the effect of depth was unusually pronounced for downstream performance.

Raffel et al. (2019) studied the effect of pretraining objectives, model structures (e.g., encoder-decoder, decoder-only), pre-training dataset size and training strategy on the transfer learning. They showed that the downstream performance monotonically increases with the model scale (from 60M to 11B parameters). While they studied several model structures, the Transformer implementation is mostly the same as the original Transformer by Vaswani et al. (2017). Conneau et al. (2020); Goyal et al. (2021) scaled-up multilingual encoder-only architectures up to 11B parameters while maintaining the original Transformer implementation. They found that scaling the model improves its cross-lingual ability. Fedus et al. (2021) scaled a sparse model based on Mixture of Experts (MoE) models up to trillion parameters.

While previous studies have repeatedly shown the benefits of scale for language understanding tasks for both dense and sparse Transformers and cross-lingual abilities, all of these used the same Transformer implementation within each studies. With a plethora of improved Transformer architectures proposed in the literature, it is timely to investigate which of these improved architecture has the best scaling properties. The main goal of this paper is to systematically study how inductive biases imposed by these Transformer variants affect the scaling behavior in a shared software and hardware settings. This is in similar spirit to (Narang et al., 2021) that studies the impact of architectures on performance. Our analysis extends that of (Narang et al., 2021) to the model scale axis.

We note that increasingly the number of tokens seen during pretraining has been incorporated in the study of scaling laws (Hoffmann et al., 2022; Muennighoff et al., 2023). Hoffmann et al. (2022) trains decoder-only Transformer language models with casual langauge modeling and evaluates on zero and few-shot tasks. In this work we consider architectures modifications that do not necessar-

ily support causal masking and autoregressive decoding. Due to this, we consider encoder-decoder configurations trained with span corruption, and evaluate on downstream finetuned tasks. This creates a more level playing field for architectures that do not support in-context learning. As such we follow (Raffel et al., 2019) to fix the number of pretraining tokens (i.e. sequence length, training steps, batch size) seen by each model. Given the large space of model architectures and scales we aim to study, this also fixes the data size dimension, making our empirical study more tractable. Since we finetune models until convergence, we anticipate the effect of pretraining token amount to be less pronounced than studied in (Hoffmann et al., 2022).

## 3 Methods

This section outlines our experimental setup.

### 3.1 Models

This section describes the models we evaluate in our experiments. Our models are largely implemented in a sequence to sequence framework (Sutskever et al., 2014) following the convention of T5 (Raffel et al., 2019). Encoder-decoder models are a natural choice for this experimentation because they can universally express both encoding and decoding tasks.

**Transformer Variants** We consider several standard Transformer variants.

- **Transformers** (Vaswani et al., 2017) - The basic vanilla Transformer architecture. Our basic setup considers the T5-style of Transformers (Raffel et al., 2019), which largely follows the vanilla Transformer except that it uses relative attention instead of sinusoidal position embeddings and pre-layer normalization, i.e. layer normalization is applied *before* each sublayer.

- **Evolved Transformers** (So et al., 2019) - A transformer architecture learned via AutoML. The architecture comprises of convolutions and attention. We scale Evolved Transformers following the same pattern as vanilla Transformers.

- **Universal Transformers (UT)** (Dehghani et al., 2018) - A Transformer architecture with shared parameters and recurrent-like computation for transform layers. Scaling UTs are challenging because of parameter sharing. While we are able

to also increase $d_{FF}$ or $d_{model}$, the increase in parameters is of magnitude $N_{layers}$ than standard Transformers. Another axis of exploration is to scale $r$ the number of repeated computation at each UT layer - this increases computation (number of FLOPs) but does not increase the parameter size of the model.

- **Switch Transformer** (Fedus et al., 2021) - a sparsely activated mixture-of-experts architecture. The Sparse Transformer is another model with an unusual relationship between number of parameters and compute. When we scale this model uniformly, the number of parameters easily reaches the ballpark of 40B.

**Efficient Transformer Variants**   These class of models are mainly concerned at reducing computational costs, memory usage, or parameter count of models.

- **Performer** (Choromanski et al., 2020) - A linear time attention model using generalizable kernel attention. For simplicity, we adopt the relu kernel variant for our experiments. We scale Performer in the similar fashion (i.e., uniform scaling) as vanilla Transformers.

- **Funnel Transformer (FT)** (Dai et al., 2020) A Transformer architecture that downsamples the input sequence across the layer stack. Our implementation uses FT only in the encoder and reverts to vanilla Transformer in the decoder following Narang et al. (2021).

- **ALBERT** (Lan et al., 2019) - A lightweight transformer architecture that shares parameters across all layers and factorizes the embedding and output softmax layers. For our seq2seq ALBERT, we also share the weights of encoder and decoder.

**General Improvements**   We consider general improvements that are not necessarily tied to Transformers. We select candidates that have shown to do well in Narang et al. (2021).

- **Mixture of Softmaxes** (Yang et al., 2017) - A transformer architecture adopting the MoS method at the Softmax layer.

- **Gated Linear Units with GeLU** (GLU-Transformer) - Replacing position-wise feed-forward-networks in Transformers with Gated Linear Units (Dauphin et al., 2017).

**Non-Transformer Architectures**   We are interested in the scaling behaviour of non-Transformer based architectures such as convolutions and/or mixer architectures.

- **Lightweight Convolutions** (Wu et al., 2019) - Lightweight depthwise convolutions that have shown promise over Transformer architectures.

- **Dynamic Convolutions** (Wu et al., 2019) - An extension of the Lightweight Convolution to create time-dependent kernels.

- **MLP-Mixers** (Tolstikhin et al., 2021) - Mixers are recently proposed architectures that learn a lightweight mixing of tokens. Since Mixers have not been used in autoregressive decoding, we only use token-mixers on the input encoder.

## 4   Experiment Setup

Our setup, along with all models, are implemented in Mesh TensorFlow (Shazeer et al., 2018), a library with similar interface to TensorFlow but enables distributed model parallelism across multiple workers. For fair comparison, all models are pre-trained for $2^{19}$ steps on the english C4 corpus optimized using an inverse square root learning rate with Adafactor (Shazeer and Stern, 2018). All models use the same SentencePiece tokenizer (Kudo and Richardson, 2018) containing $32K$ subwords. This closely follows the setup in the T5 paper (Raffel et al., 2019). Finetuning is performed for $100K$ steps on a mixture of GLUE (Wang et al., 2018), SuperGLUE (Wang et al., 2019) and SQuAD (Rajpurkar et al., 2016). We evaluate on both upstream (pre-training) validation perplexity as well as downstream transfer for NLU tasks (GLUE + SuperGLUE + SQuAD) after fine-tuning. We pretrain and finetune our models with 16 TPU-v3 chips with data parallelism. All large models have a model parallelism of 2 and XL models have a model parallelism of 8.

**Model Sizes**   We consider several different model sizes for each architecture. For models that are straightforward to scale, we simply follow the standard convention in Raffel et al. (2019), moving from small to base, to large and XL. We include a tiny version of each model to observe how different models behave at lower compute regions. For models where it was not straightforward to scale (e.g., Universal Transformers, ALBERT), we tried

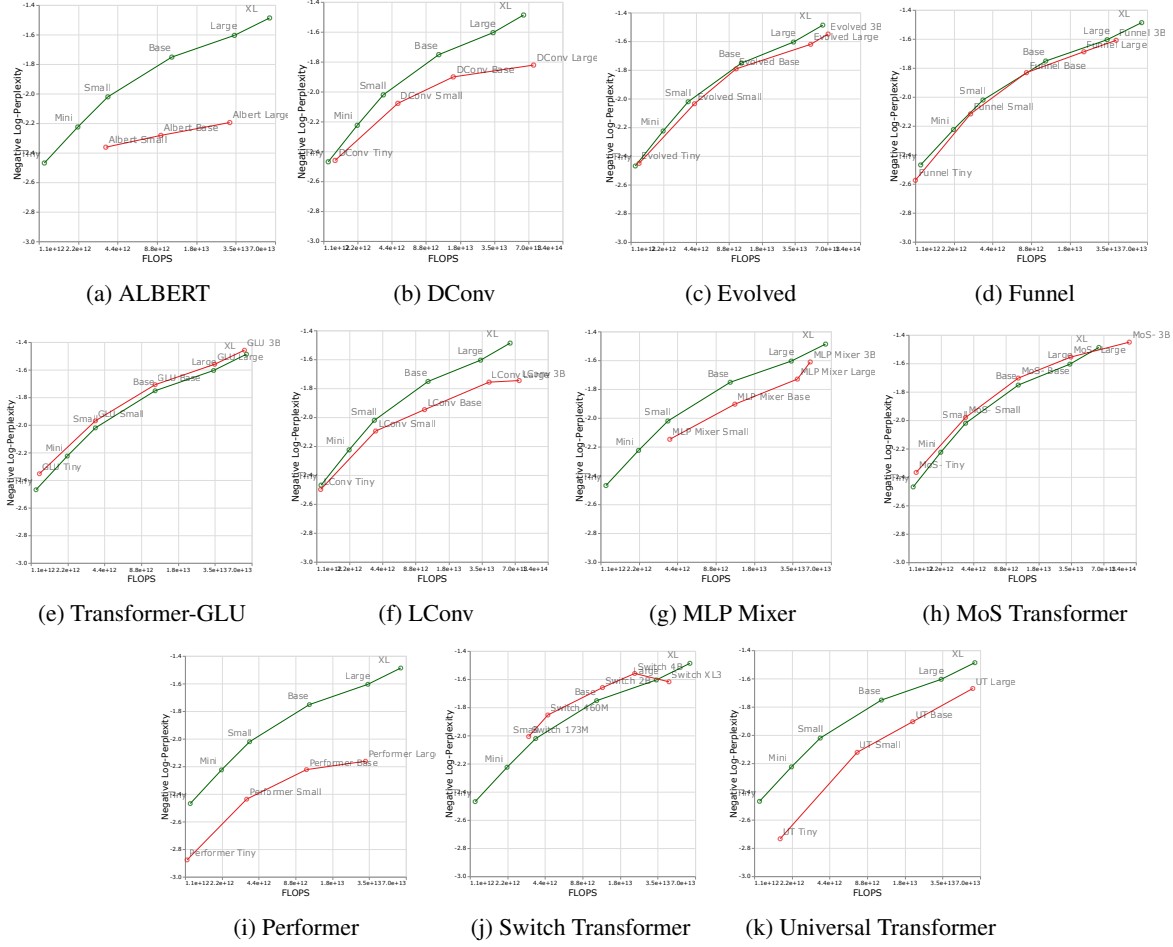

| (a) ALBERT | (b) DConv | (c) Evolved | (d) Funnel |

| (e) Transformer-GLU | (f) LConv | (g) MLP Mixer | (h) MoS Transformer |

| (i) Performer | (j) Switch Transformer | (k) Universal Transformer |

Figure 2: Upstream Negative Log-Perplexity of vanilla Transformer compared to other models.

to scale them in a similar fashion but faced obvious limitations such as getting ALBERT to have the same number of parameters as T5 XL without incurring a huge number of cost in terms of FLOPs. For convolutional models, we consider $d_{\mathrm{model}}$ to be the hidden size (i.e., channel depth) for the one-dimensional convolution layers. Values such as $d_{\mathrm{kv}}, N_H$ then become redundant. Details on scaling details[1] of each architecture can be found in the supplementary material.

## 5 Main Results

We report the main results of this paper in Table 1. We report the number of trainable parameters, FLOPs (of a single forward pass) and speed (steps per second). We also report on validation perplexity (on upstream pre-training) and results on 17 downstream tasks. The results are reported aggre-

gates of GLUE, SuperGLUE and SQuAD. While we use the same Mesh TensorFlow-based codebase used by Raffel et al. (2019) and hence expect our experimental results to match theirs, we verify that our T5 base does achieve similar results to what is reported in Raffel et al. (2019).

### 5.1 Do all models scale the same way?

We compare on both upstream perplexity and downstream finetuning performance here.

**Upstream Perplexity** Figure 2 reports the scaling behaviour of all models as we increase the number of FLOPs. We observe that the scaling behaviour of all models are quite unique and distinct, i.e., most of them are quite different from standard Transformers. Perhaps the biggest finding here is that most models (e.g., LConv, Evolved) all seem to be on-par or better than standard Transformers but fail to scale with a higher compute budget. Another interesting trend is that "linear" Transformers such as Performer fail to scale as shown in Figure 2i.

---

[1]The largest Switch transformer was scaled in a pretty sub-optimal way. So we don't think it is representative of the full potential of the Switch family. Take the last data point of Switch with a pinch of salt.

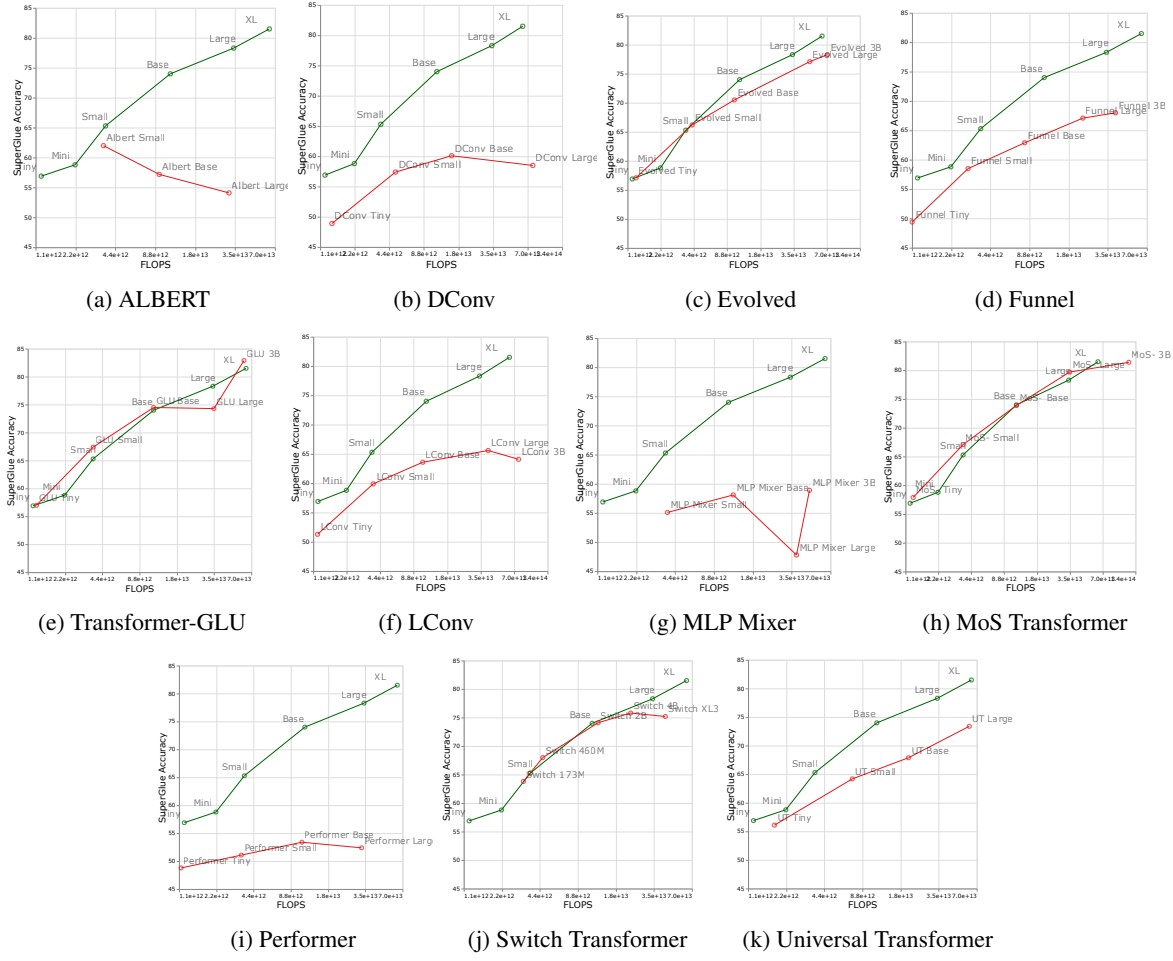

Figure 3: Downstream accuracy of vanilla Transformer compared to other models.

The pre-training perplexity metric only decreases by 2.7% going from base to large scale compared to 8.4% of the vanilla Transformer.

**Downstream Transfer** Figure 3 reports the scaling curves of all models on downstream transfer. The overall finding that most models have distinct scaling curves compared to Transformers is also evident in downstream tasks. It is also noteworthy that most models have a different upstream and downstream scaling curve. We find that some models such as Funnel Transformer and LConvs that seem to hold out pretty well on upstream but suffer substantially on downstream. As for Performer, the performance (disparity) seems to be even greater in downstream as compared to upstream. Notably, the SuperGLUE downstream tasks generally require pseudo cross-attention on the encoder, which models such as convolutions are not equipped to handle (Tay et al., 2021a). To this end, we find that certain models may have difficulty learning the downstream tasks despite good upstream performance.

## 5.2 Are the best models at each scale different?

Figure 1 shows the Pareto-frontier when plotting compute against upstream and downstream performance. Since the colors of the plot represent different models, we can observe that the best model for every scale and compute region might be different. Moreover, from Figure 3, we can also observe this. For example, the Evolved Transformer seems to do well against the standard Transformer at tiny to small region (downstream) but this quickly changes when scaling the model up. We also observe this with MoS-Transformer where it clearly outperforms vanilla Transformers at some regions but not at others.

## 5.3 Scaling Law for Each Model

Table 2 presents the slope of the fitted linear line $\alpha$ for each model across multiple scenarios. We derive $\alpha$ by plotting $F$ (FLOPs), $U$ (upstream perplexity), $D$ (downstream accuracy), $P$ (number of parameters). In general, most values of $\alpha$ depict

Table 1: Results on pre-training and finetuning ten different model architectures. Full results (further varying hyperparameters of these models) can be found in the Appendix.

| Model | #Params | FLOPs | Speed | Neg Log Ppl | GLUE | SGLUE | SQuAD |
|---|---|---|---|---|---|---|---|
| Transformer Tiny | 16M | 1.21 | 38.4 | -2.47 | 69.3 | 56.9 | 73.6 |
| Transformer Small | 60M | 3.70 | 22.7 | -2.02 | 78.1 | 65.3 | 81.9 |
| Transformer Base | 223M | 11.4 | 9.3 | -1.75 | 83.8 | 74.0 | 86.3 |
| Transformer Large | 738M | 34.3 | 3.6 | -1.61 | 86.4 | 78.3 | 88.6 |
| Transformer XL | 2.9B | 63.8 | 1.3 | -1.49 | 87.8 | 81.5 | 89.5 |
| Evolved Transformer Tiny | 19M | 1.31 | 39.7 | -2.45 | 69.6 | 57.1 | 69.6 |
| Evolved Transformer Small | 79M | 4.23 | 23.7 | -2.04 | 75.7 | 66.2 | 80.2 |
| Evolved Transformer Base | 218M | 10.2 | 8.9 | -1.79 | 83.0 | 70.5 | 84.8 |
| Evolved Transformer Large | 1.0B | 49.3 | 2.1 | -1.62 | 86.2 | 77.1 | 88.0 |
| Evolved Transformer XL | 2.2B | 71.3 | 0.8 | -1.55 | 87.0 | 78.3 | 88.2 |
| Universal Transformer Tiny | 11M | 1.77 | 38.1 | -2.73 | 69.8 | 56.1 | 62.3 |
| Universal Transformer Small | 52M | 7.30 | 18.3 | -2.12 | 76.8 | 64.2 | 75.4 |
| Universal Transformer Base | 127M | 20.3 | 8.4 | -1.91 | 80.0 | 67.9 | 80.1 |
| Universal Transformer Large | 283M | 27.6 | 1.6 | -1.67 | 84.0 | 73.4 | 85.4 |
| Switch Transformer Tiny | 174M | 3.25 | 29.7 | -2.01 | 78.2 | 63.8 | 80.7 |
| Switch Transformer Small | 460M | 4.63 | 22.3 | -1.85 | 80.3 | 68.0 | 82.9 |
| Switch Transformer Base | 2.0B | 12.7 | 8.4 | -1.66 | 84.2 | 74.1 | 86.5 |
| Switch Transformer Large | 3.9B | 23.0 | 4.1 | -1.56 | 84.6 | 75.8 | 87.9 |
| Switch Transformer XL | 29.6B | 43.3 | 0.8 | -1.62 | 84.0 | 75.2 | 87.5 |
| Performer Tiny | 16M | 1.14 | 42.0 | -2.88 | 50.5 | 48.8 | 15.0 |
| Performer Small | 61M | 3.50 | 39.0 | -2.44 | 57.8 | 51.1 | 31.1 |
| Performer Base | 224M | 10.8 | 11.7 | -2.23 | 61.4 | 53.4 | 37.8 |
| Performer Large | 739M | 32.8 | 4.4 | -2.16 | 62.4 | 52.4 | 30.8 |
| Funnel Transformer Tiny | 16M | 1.10 | 39.9 | -2.58 | 63.4 | 49.4 | 54.6 |
| Funnel Transformer Small | 61M | 2.96 | 32.7 | -2.11 | 70.0 | 58.5 | 75.1 |
| Funnel Transformer Base | 223M | 8.10 | 11.9 | -1.83 | 76.3 | 62.9 | 81.6 |
| Funnel Transformer Large | 739M | 22.6 | 5.0 | -1.69 | 79.8 | 67.1 | 83.8 |
| Funnel Transformer XL | 2.9B | 40.3 | 1.89 | -1.61 | 79.8 | 68.0 | 83.7 |
| ALBERT Small | 15M | 3.57 | 42.0 | -2.36 | 73.7 | 62.0 | 77.1 |
| ALBERT Base | 21M | 9.40 | 16.4 | -2.28 | 69.0 | 57.2 | 64.3 |
| ALBERT Large | 34M | 31.6 | 5.1 | -2.20 | 62.9 | 54.1 | 27.3 |
| MoS-Transformer Tiny | 27M | 1.29 | 39.7 | -2.37 | 70.6 | 57.9 | 74.1 |
| MoS-Transformer Small | 81M | 3.70 | 26.3 | -1.98 | 79.7 | 67.1 | 83.1 |
| MoS-Transformer Base | 257M | 11.4 | 8.6 | -1.70 | 84.5 | 73.9 | 86.8 |
| MoS-Transformer Large | 800M | 35.0 | 3.4 | -1.56 | 86.5 | 79.7 | 89.1 |
| MoS-Transformer XL | 2.9B | 112 | 1.2 | -1.45 | 88.2 | 81.4 | 90.0 |
| GLU-Transformer Tiny | 26M | 1.29 | 31.7 | -2.35 | 70.5 | 57.0 | 74.2 |
| GLU-Transformer Small | 77M | 3.70 | 26.4 | -1.97 | 79.1 | 67.4 | 83.0 |
| GLU-Transformer Base | 248M | 11.4 | 8.6 | -1.71 | 84.6 | 74.5 | 87.2 |
| GLU-Transformer Large | 748M | 35.0 | 3.4 | -1.56 | 84.2 | 74.3 | 86.2 |
| GLU-Transformer XL | 2.85B | 61.3 | 1.0 | -1.49 | 87.6 | 82.9 | 89.4 |
| LConv Tiny | 17M | 1.20 | 31.2 | -2.50 | 51.1 | 51.3 | 49.5 |
| LConv Small | 67M | 3.80 | 12.8 | -2.10 | 71.8 | 59.9 | 64.7 |
| LConv Base | 210M | 10.6 | 12.8 | -1.95 | 73.8 | 63.6 | 70.3 |
| LConv Large | 741M | 41.0 | 3.0 | -1.76 | 76.8 | 65.6 | 76.3 |
| LConv XL | 2.3B | 77.0 | 1.0 | -1.75 | 73.3 | 64.1 | 72.9 |
| DConv Tiny | 22M | 1.39 | 27.3 | -2.46 | 51.1 | 48.9 | 30.2 |
| DConv Small | 96M | 4.97 | 19.8 | -2.08 | 68.6 | 57.4 | 64.3 |
| DConv Base | 324M | 15.3 | 7.6 | -1.90 | 72.9 | 60.1 | 63.7 |
| DConv Large | 1.2B | 78.0 | 1.1 | -1.82 | 70.8 | 58.5 | 58.2 |
| MLP-Mixer Small | 67M | 3.83 | 22.3 | -2.15 | 65.4 | 55.1 | 58.7 |
| MLP-Mixer Base | 233M | 12.4 | 10.7 | -1.90 | 64.4 | 58.1 | 60.5 |
| MLP-Mixer Large | 739M | 38.3 | 3.9 | -1.73 | 52.2 | 47.8 | 60.9 |
| MLP-Mixer XL | 2.86B | 48.3 | 1.2 | -1.61 | 57.3 | 58.9 | 65.7 |

Table 2: Slope of a fitted linear line for each model, when we compare FLOPs vs. upstream performance $(F, U)$, FLOPs vs. downstream performance $(F, D)$, parameter size vs. upstream performance $(F, U)$, parameter size vs. downstream performance $(P, D)$, and finally upstream vs. downstream performance $(U, D)$.

| Model | $\alpha_{F,U}$ | $\alpha_{F,D}$ | $\alpha_{P,U}$ | $\alpha_{P,D}$ | $\alpha_{U,D}$ |
|---|---|---|---|---|---|
| Transformer | **0.54** | **0.28** | **0.47** | **0.24** | 0.49 |
| GLU-Trans. | 0.49 | 0.24 | 0.42 | 0.22 | 0.46 |
| LConv | 0.32 | 0.13 | 0.29 | 0.11 | 0.48 |
| Funnel | 0.47 | 0.22 | 0.38 | 0.18 | 0.46 |
| Switch | 0.23 | 0.14 | 0.13 | 0.08 | **0.58** |
| Universal | 0.50 | 0.20 | 0.56 | 0.22 | 0.35 |
| ALBERT | 0.08 | -0.12 | 0.13 | -0.21 | -1.67 |
| Evolved | 0.44 | 0.22 | 0.42 | 0.21 | 0.47 |
| Performer | 0.25 | 0.05 | 0.24 | 0.05 | 0.24 |
| MoS-Trans. | 0.43 | 0.21 | 0.43 | 0.20 | 0.47 |
| MLP-Mixer | 0.32 | -0.03 | 0.26 | 0.65 | -0.02 |

how well a model scales. For example $\alpha_{F,U}$ is plotting FLOPs against Upstream performance. The only exception is $\alpha_{U,D}$ which is a measure of upstream vs downstream performance. A high $\alpha_{U,D}$ value means that the transfer to the downstream tasks is better as a model scales. Overall, the $\alpha$ value is a metric that represents how well a model performs relatively across all scales

**Analysis of Slope for each Model** In general, we find that the vanilla Transformer has the highest values of $\alpha$. Models such as Evolved Transformer, GLU-Transformer, MoS-Transformer and Funnel Transformer tend to have similar scaling properties to the vanilla Transformer. The GLU-Transformer has similar and slightly worse scaling properties to the vanilla Transformer, even if it was observed to do better in absolute sense on some compute-regions. On the other hand, we also observe that there are models which are difficult to scale such as LConv, UT, MLP-Mixer and Performer. This is even more evident on downstream task. We also note that ALBERT scales (trends) negatively[2] (gets worse) as we scale the model up. On the other hand, the metric $\alpha_{U,D}$ measures how the downstream performance scales with upstream performance. Overall, the Switch Transformer does the best on this metric where downstream performance scales well with upstream performance. Generally, models that make less changes to the main Transformer architecture (GLU-Transformer, MoS-Transformer) tend to retain similar scaling behaviours and changing the inductive bias also significantly alters the

---

[2]This version of ALBERT shares parameters across encoder and decoder which may partially explain why we had a hard time scaling up.

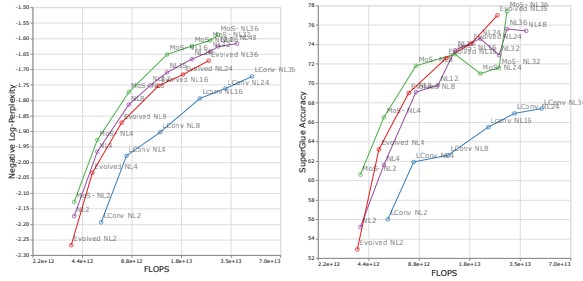

(a) Upstream Neg. Log-PPL. (b) Downstream Accuracy.

Figure 4: Scaling depth

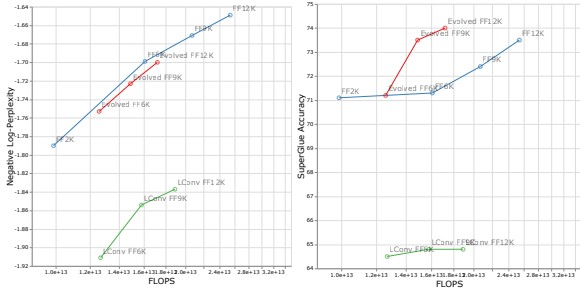

(a) Upstream Neg. Log-PPL. (b) Downstream Accuracy.

Figure 5: Scaling width of FFN

scaling property of the model.

### 5.4 Do Scaling Protocols influence model architectures in the same way?

We are interested in how different scaling protocols influence the model architectures. Figure 4 shows the effect of scaling depth of four model architectures (MoS-Transformer, Transformer, Evolved Transformer and LConv). Figure 5 shows the effect of scaling width on the same four architectures. Firstly, on upstream (negative log perplexity) curves, we note that while different architectures have a distinct difference in absolute performance, the scaling trend remains quite similar. On downstream, depth scaling (Figure 4) seems to act equally on most architectures with the exception of LConv. Meanwhile, for width scaling, it seems that Evolved Transformers scale slightly better when applying width-scaling. It is also interesting to note that depth-scaling has a much more substantial impact on downstream scaling as opposed to width-scaling.

### 6 Epilogue and Conclusion

In this paper, we conducted extensive experiments, pretraining and finetuning of up to 100 models ranging from 10 well-established Transformer and non-Transformer architectures. We showed that

different model architectures can have different scaling behaviours and models performing well in one compute region (or model size) may not do identically well in another compute region. We also showed that model architectures may do well on upstream perplexity but fail to transfer to downstream tasks. Hence, practitioners should be cautious about developing architectures that not only scale well with respect to the upstream perplexity, but also based on downstream performance. While we certainly do not expect researchers to always report model performance across all scales (especially large-scale), we believe that it is good to keep in mind that architectures can perform quite differently at different compute regions. Hence, this might be a good dimension to consider when designing new inductive biases. As such, performing evaluation at a certain compute region may be insufficient to capture the full picture. It is also good to consider if different inductive biases will result in different extents of emergent capabilities (Wei et al., 2022a; Abnar et al., 2020).

We also showed that different model architectures may react differently to different scaling protocols, reaffirming that comparing and benchmarking these models can be very challenging (Dehghani et al., 2021b). When it comes to scaling large models, we show that novel inductive biases can be indeed quite risky which might explain why most state-of-the-art large language models (Rae et al., 2021; Chowdhery et al., 2022; Tay et al., 2022) are based on relatively vanilla architectures. Our advice is to be cautious when staking an expensive run on an architecture that drastically modifies the attention mechanism. Finally, we acknowledge that not every practitioner or researcher would require models that are able to scale to billions of parameters. In that case, inductive biases that are tailored to small or low compute will be sufficient.

## 7  Limitations

As with all empirical studies, ours come with its own set of limitations. We only present a sampling of Transformer variants, and it is not exhaustive. Our selection is aimed towards sampling a diversity of architecture approaches to have representation across the entire space of Transformer architectures. As such, we do not claim that our findings hold within a subcategory; for example efficient Transformer variants, which there are many recent works not covered here. Additionally, given the huge number of models considered in this work, while we scaled each model to the best of our ability and present details on how they were scaled, there could always be unexplored hyperparameter settings and other tricks that could get a model to "work" at larger scales. Beyond this work, one could also study the differences in prompting techniques, e.g. chain-of-thought prompting (Wei et al., 2022b), between different architecture and scales. Such findings would be of importance for the research community in the future. Although in either case here, we believe that our findings, i.e. models scale differently and need to be tested, will continue to be relevant. This space will only continue to grow, and future researchers and practitioners must continue to assess the scalability of new models under new use cases.

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

# 8 Appendix

## 8.1 Scaling Details for Individual Models

For most models, it was reasonable to follow the uniform scaling method in the main T5 sizes. At each size, the hyperparameters are as follows:

| Model | $N_L$ | $d_{ff}$ | $d_{model}$ | $d_{kv}$ | $N_H$ | #Params |
|---|---|---|---|---|---|---|
| Tiny | 4/4 | 1024 | 256 | 32 | 4 | 16M |
| Small | 6/6 | 2048 | 512 | 32 | 8 | 60M |
| Base | 12/12 | 3072 | 768 | 64 | 12 | 220M |
| Large | 24/24 | 4096 | 1024 | 64 | 16 | 738M |
| XL | 24/24 | 16384 | 1024 | 128 | 32 | 3B |

Table 3: Table of model configurations. $N_L$ is the number of layers, $d_{ff}$ is the size of the MLP, $d_{model}$ is the hidden size of the model. $d_{kv}$ is the size of each key-value vector. $N_H$ is the number of heads.

**Scaling for Switch Transformer** For Switch Transformers, we use the following scaling:

| Model | $N_L$ | $d_{ff}$ | $d_{model}$ | $d_{kv}$ | $N_H$ | $N_E$ | #Params |
|---|---|---|---|---|---|---|---|
| Tiny | 4 | 1024 | 512 | 64 | 12 | 32 | 173M |
| Small | 6 | 2048 | 512 | 64 | 12 | 32 | 460M |
| Base | 12 | 3072 | 768 | 64 | 12 | 32 | 2B |
| Large | 24 | 3072 | 768 | 64 | 12 | 32 | 8B |
| XL | 48 | 3072 | 768 | 64 | 12 | 128 | 30B |

Table 4: Scaling for Switch Transformer. $N_E$ is the number of experts.

**Scaling for Universal Transformer** Scaling UTs are generally difficult as described in the main text. There were two main considerations for scaling UTs. Initially we tried scaling the number of recurrent operations. However, we found that even with an increase of FLOPS, this does not lead to improved performance. Overall, the UT model might be pretty slow and therefore a model with the same hparams as vanilla XL might be infeasible to run. Hence, we explored increasing the width of the MLPs to $32K$ to see if UTs would scale in this manner.

| Model | $N_R$ | $d_{ff}$ | $d_{model}$ | $d_{kv}$ | $N_H$ | #Params |
|---|---|---|---|---|---|---|
| UT Tiny | 3/3 | 1024 | 128 | 32 | 8 | 11M |
| UT Small | 3/3 | 2048 | 512 | 32 | 8 | 52M |
| UT Base | 3/3 | 3072 | 768 | 64 | 12 | 127M |
| UT Large | 3/3 | 32768 | 1024 | 64 | 16 | 283M |

Table 5: Table of model configurations. $N_R$ is the number of recurrent operations, $d_{ff}$ is the size of the MLP, $d_{model}$ is the hidden size of the model. $d_{kv}$ is the size of each key-value vector. $N_H$ is the number of heads.

## 8.2 Full Results

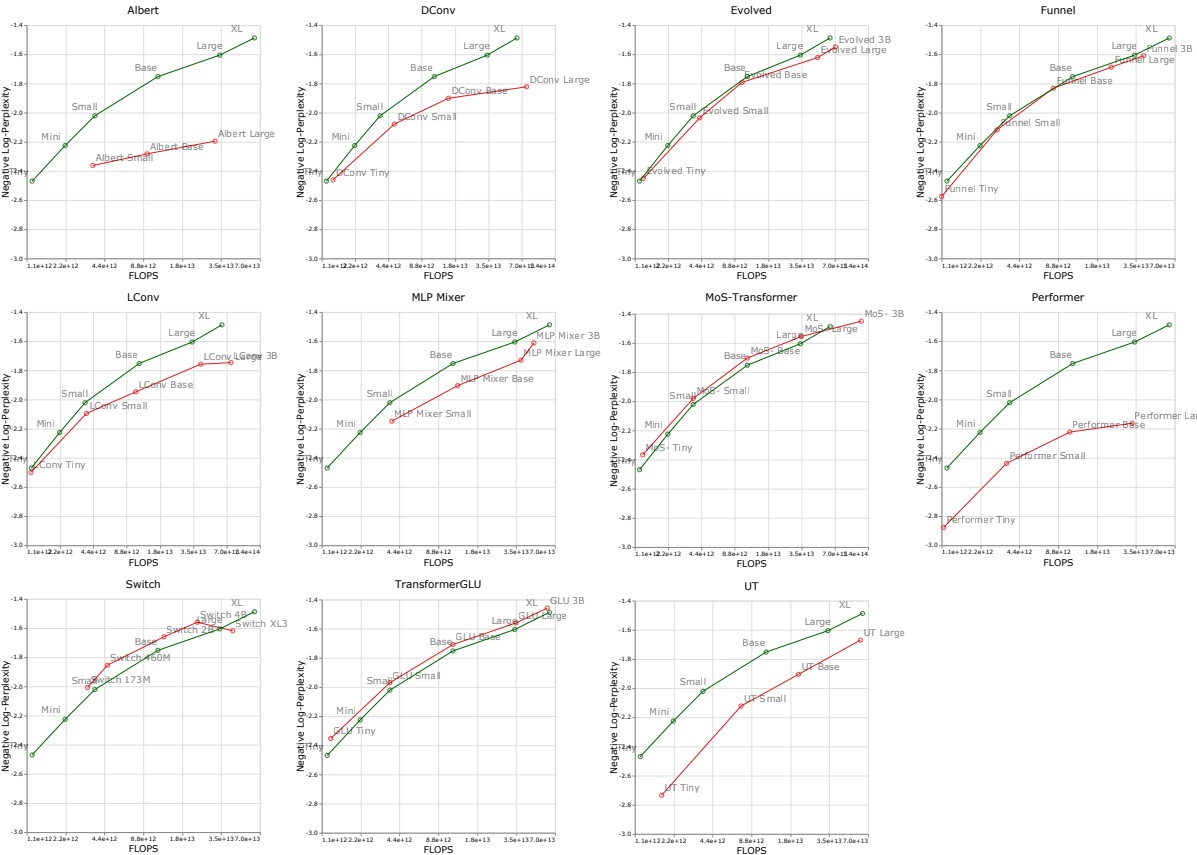

Figure 6: Quality-FLOP trade off for the upstream Negative Log-Perplexity of vanilla Transformer compared to other models.

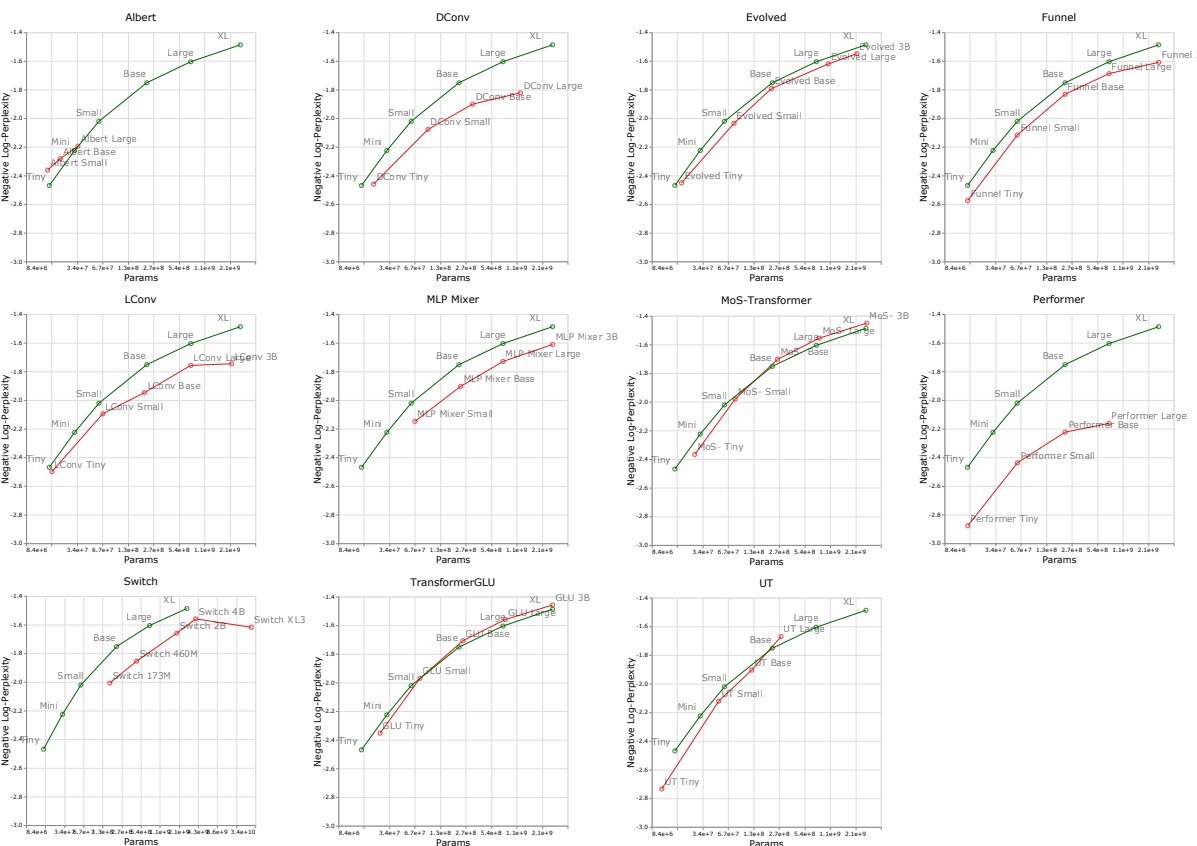

Figure 7: Quality-Parameter trade off for the upstream Negative Log-Perplexity of vanilla Transformer compared to other models.

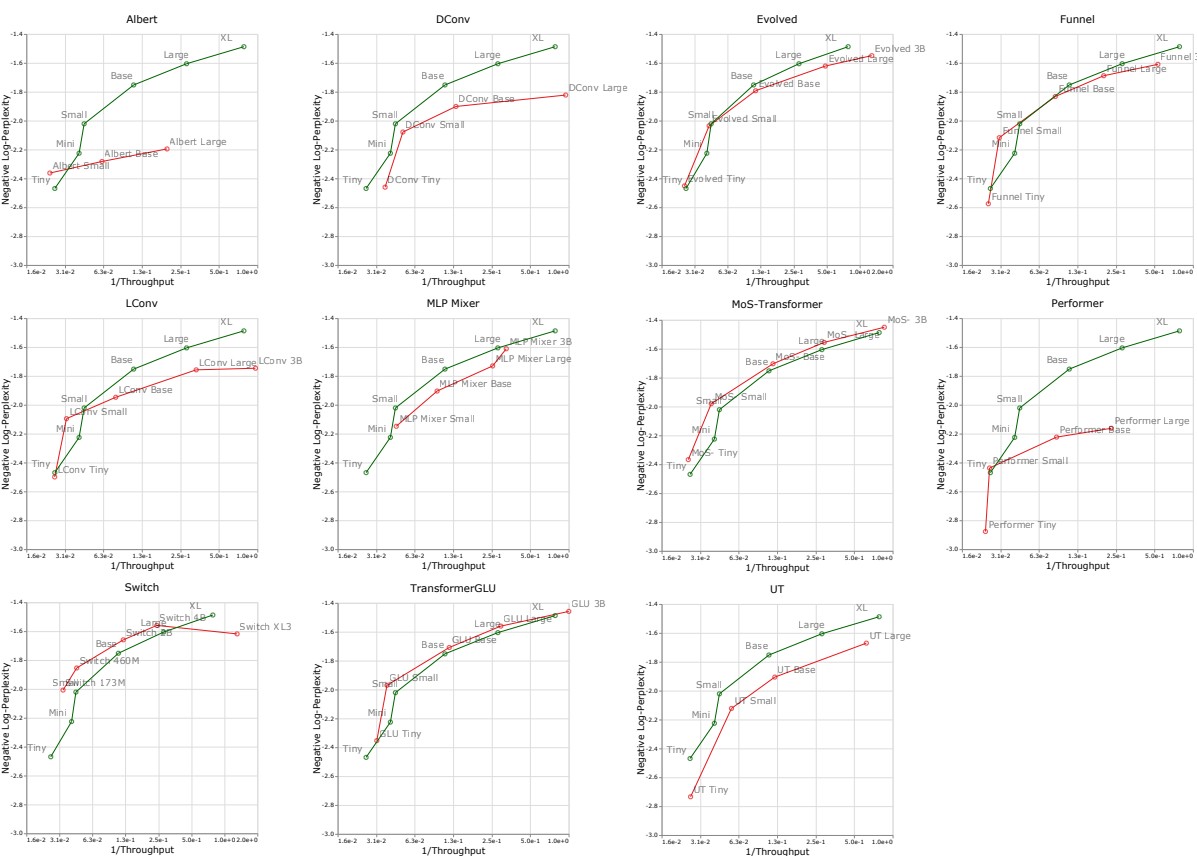

Figure 8: Quality-Throughput trade off for the upstream Negative Log-Perplexity of vanilla Transformer compared to other models.

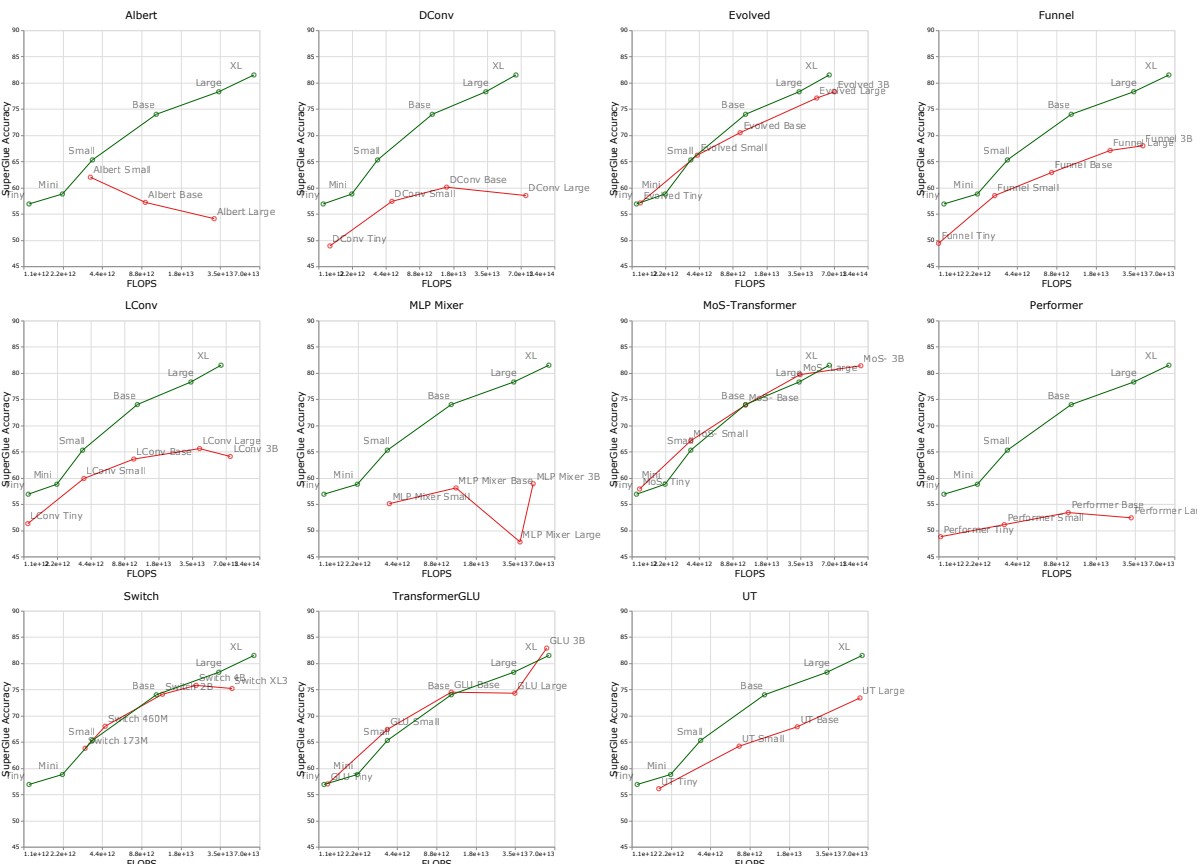

Figure 9: Quality-FLOP trade off for the downstream SuperGlue Accuracy of vanilla Transformer compared to other models, with respect to FLOPs, number of parameters, and throughput.

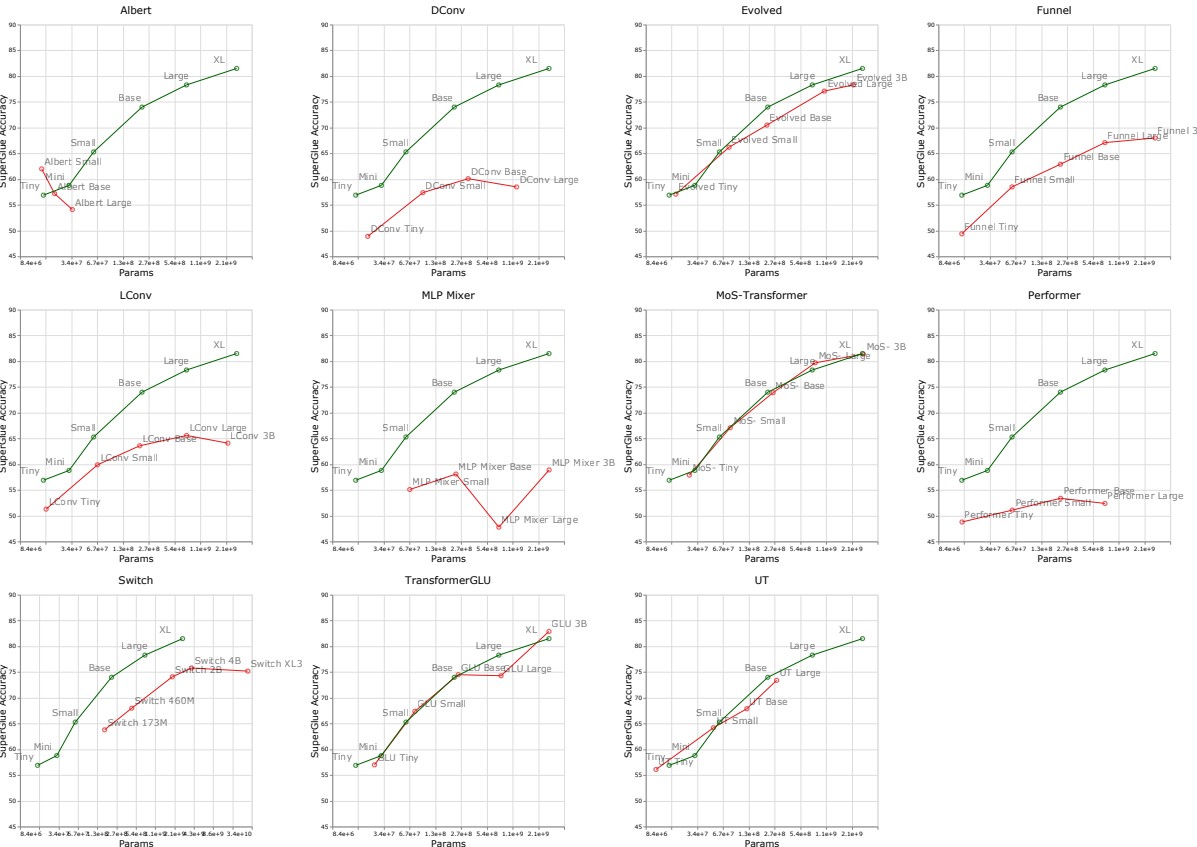

Figure 10: Quality-Parameter trade off for the downstream SuperGlue Accuracy of vanilla Transformer compared to other models.

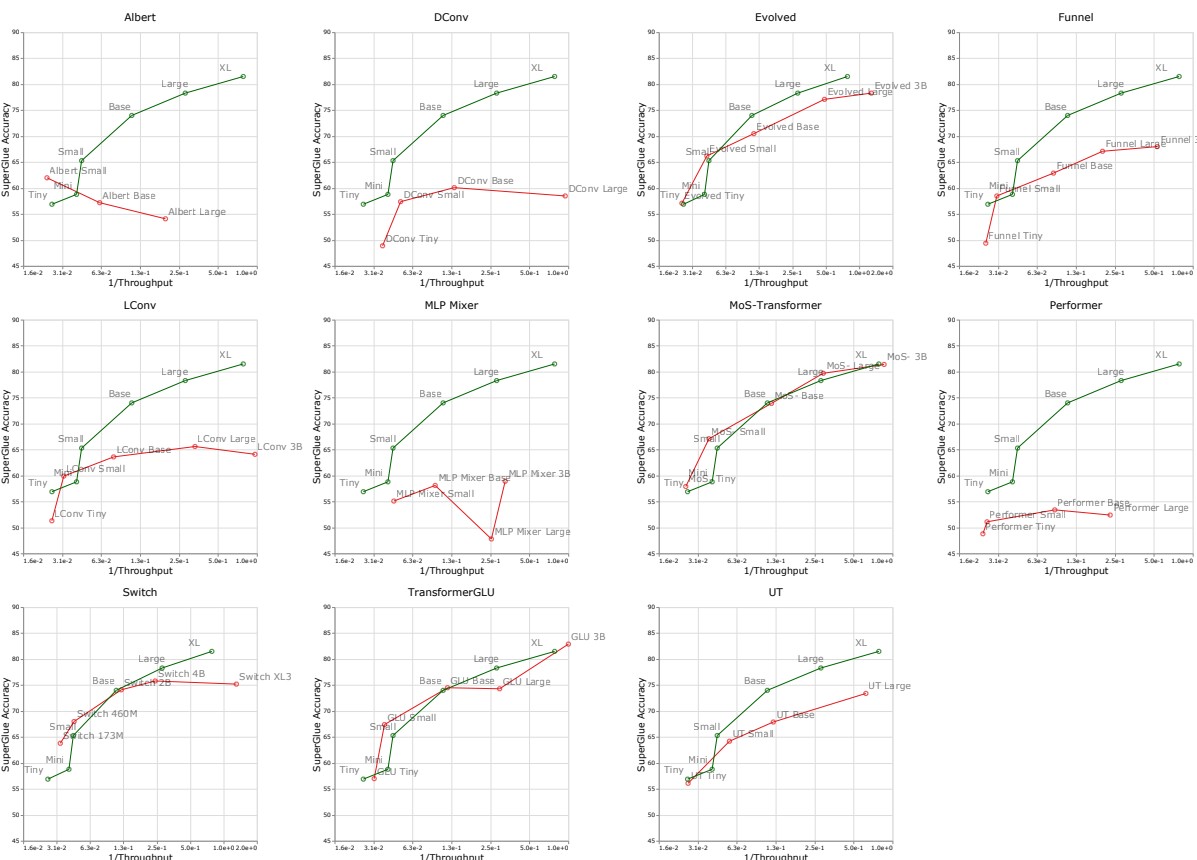

Figure 11: Quality-Throughput trade off for the downstream SuperGlue Accuracy of vanilla Transformer compared to other models.

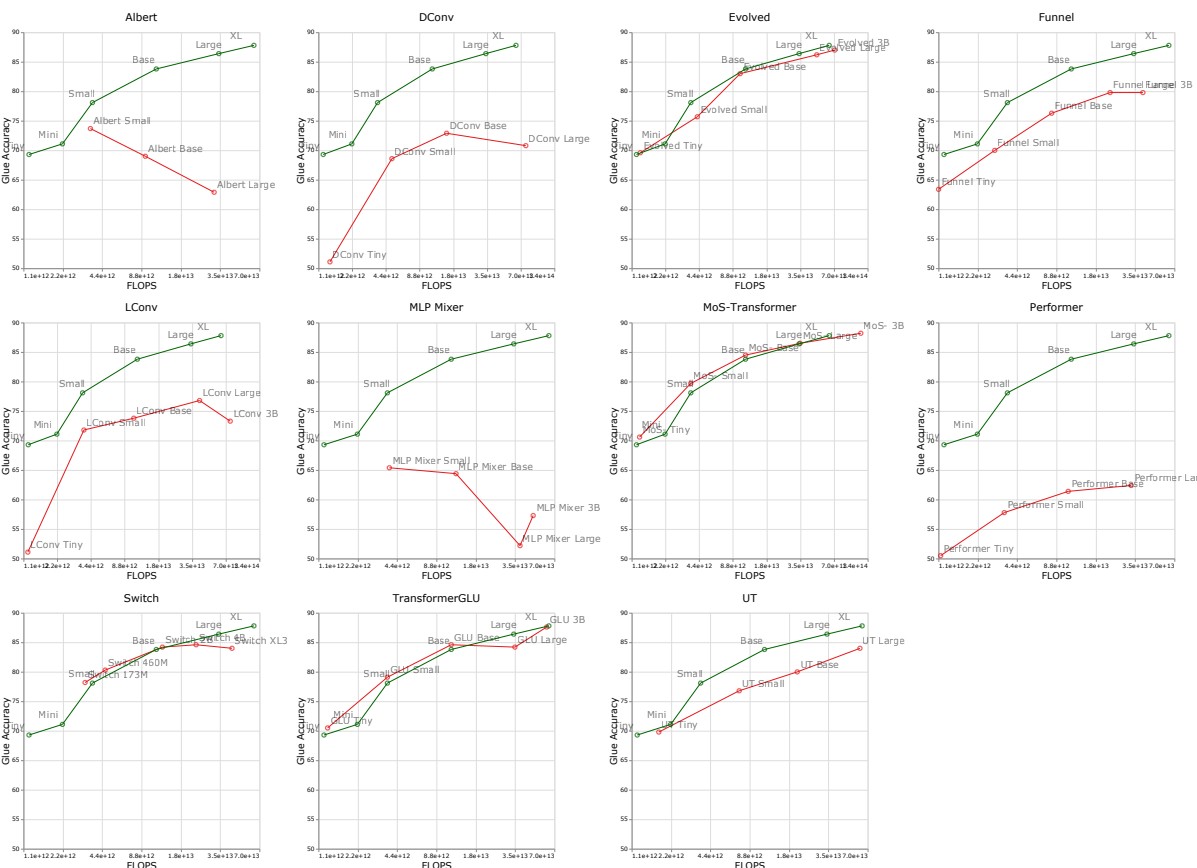

Figure 12: Quality-FLOP trade off for the downstream Glue Accuracy of vanilla Transformer compared to other models.

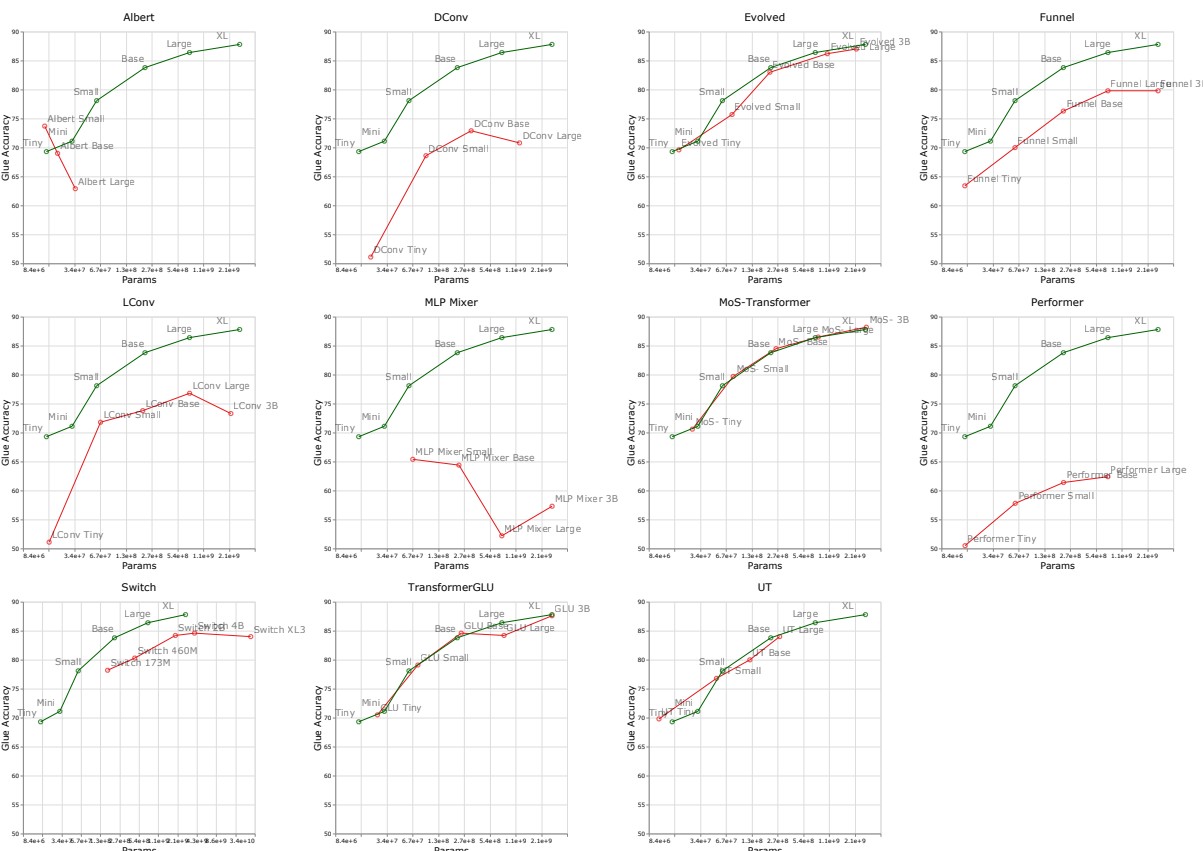

Figure 13: Quality-Parameter trade off for the downstream Glue Accuracy of vanilla Transformer compared to other models.

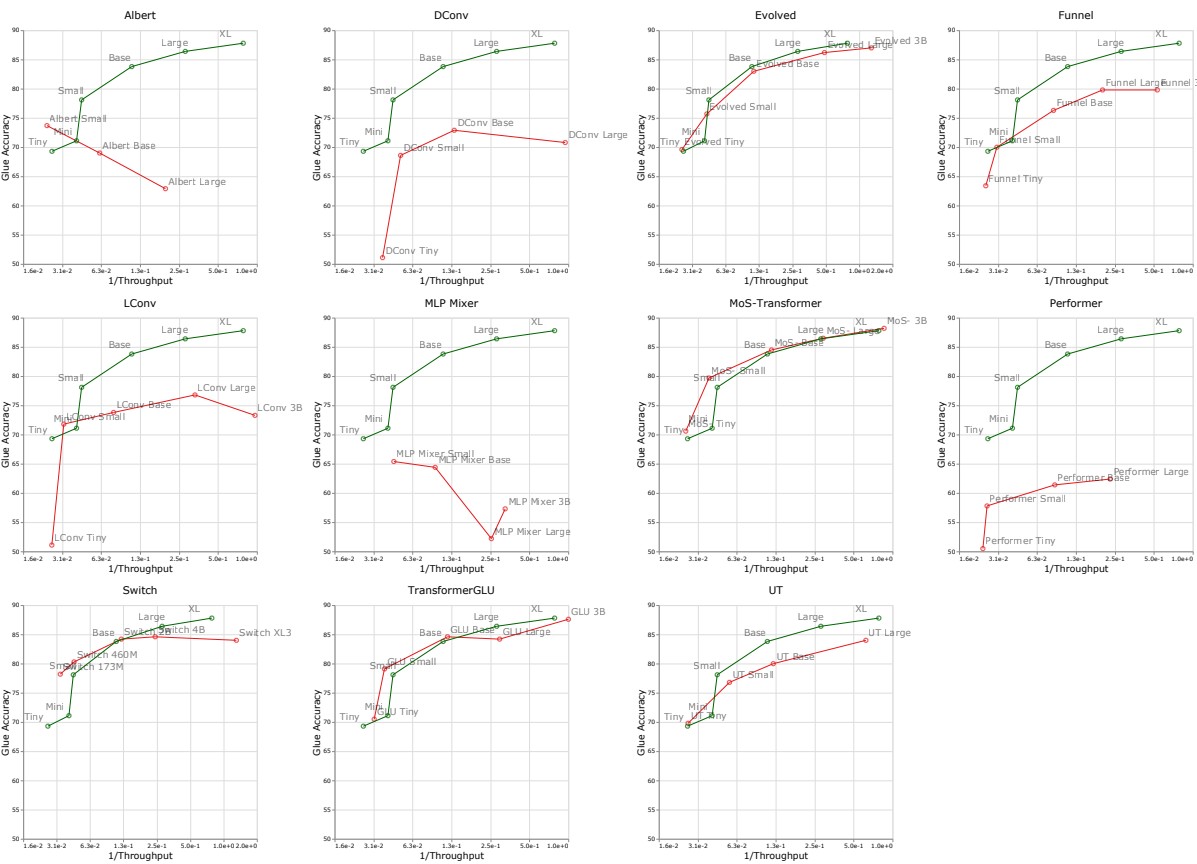

Figure 14: Quality-Throughput trade off for the downstream Glue Accuracy of vanilla Transformer compared to other models.

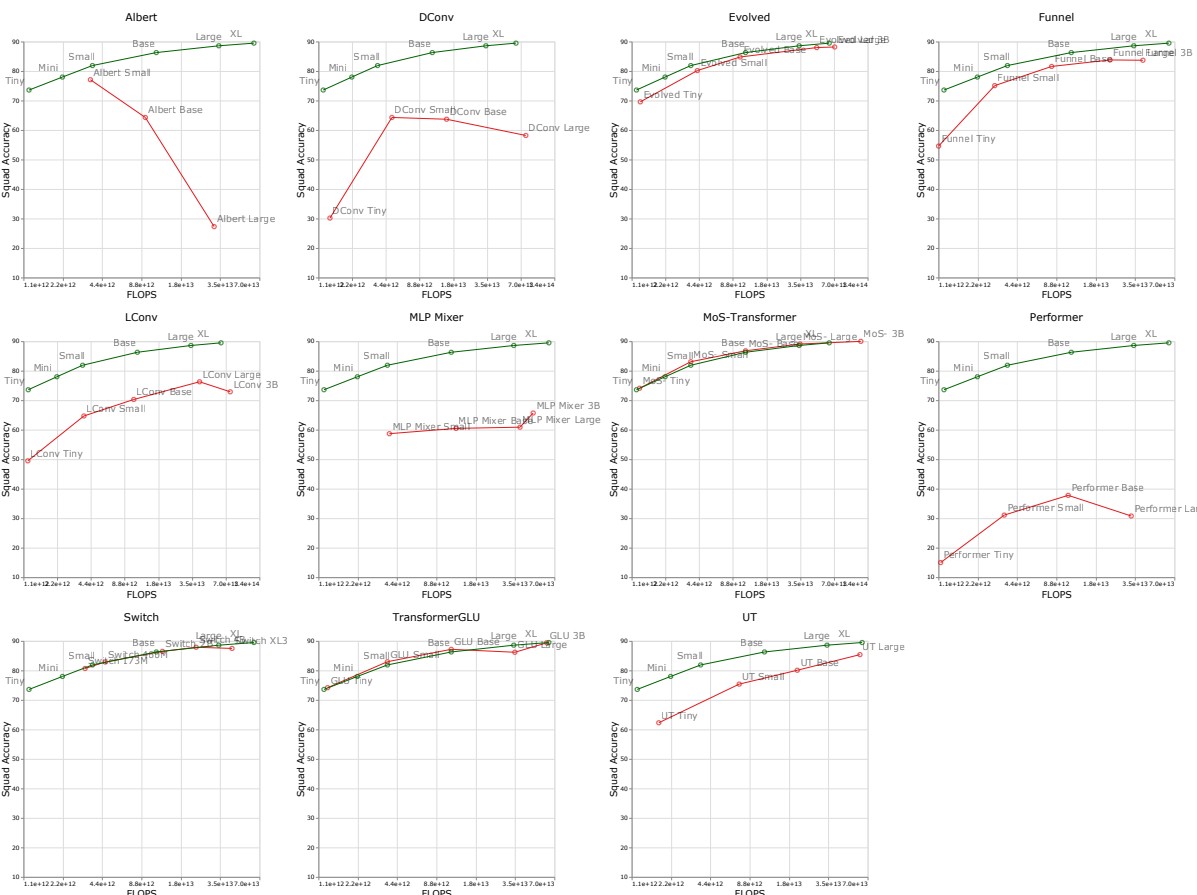

Figure 15: Quality-FLOP trade off for the downstream Squad Accuracy of vanilla Transformer compared to other models.

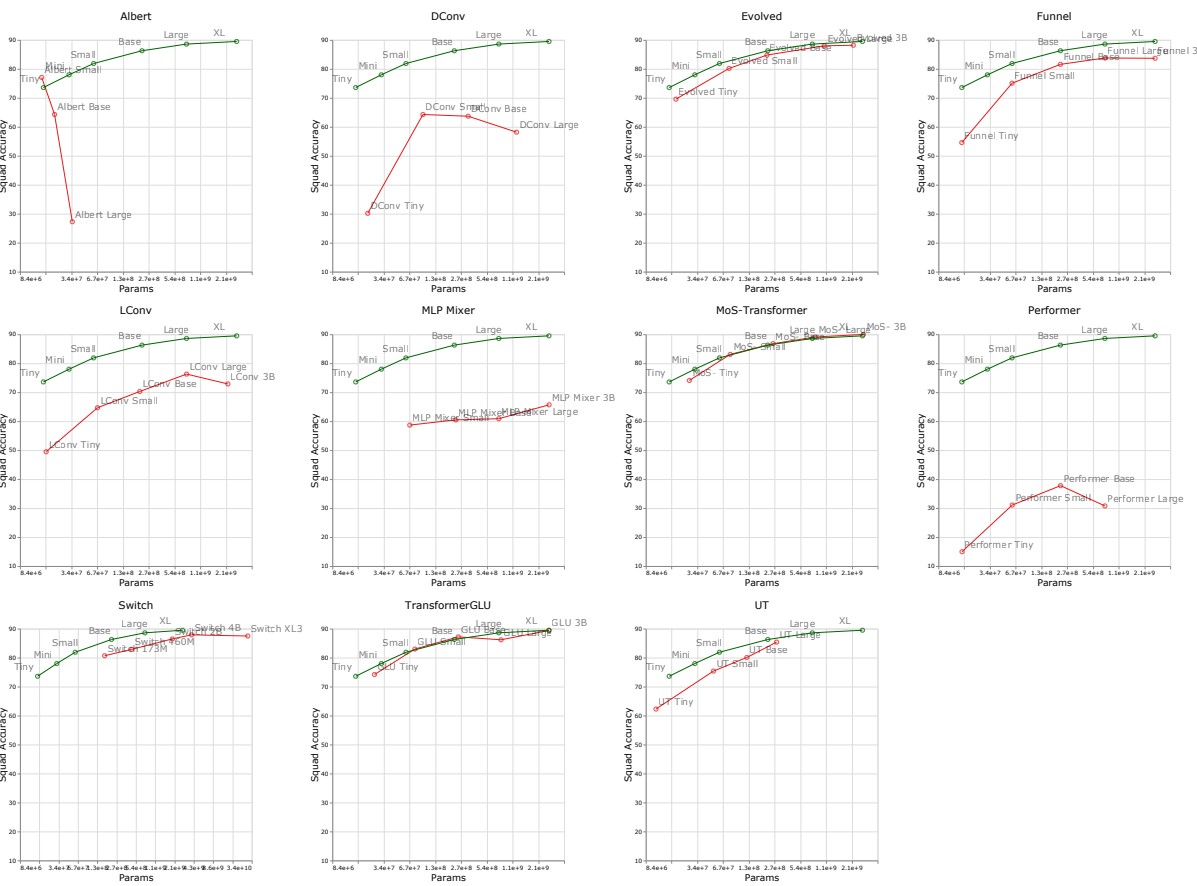

Figure 16: Quality-Parameter trade off for the downstream Squad Accuracy of vanilla Transformer compared to other models.

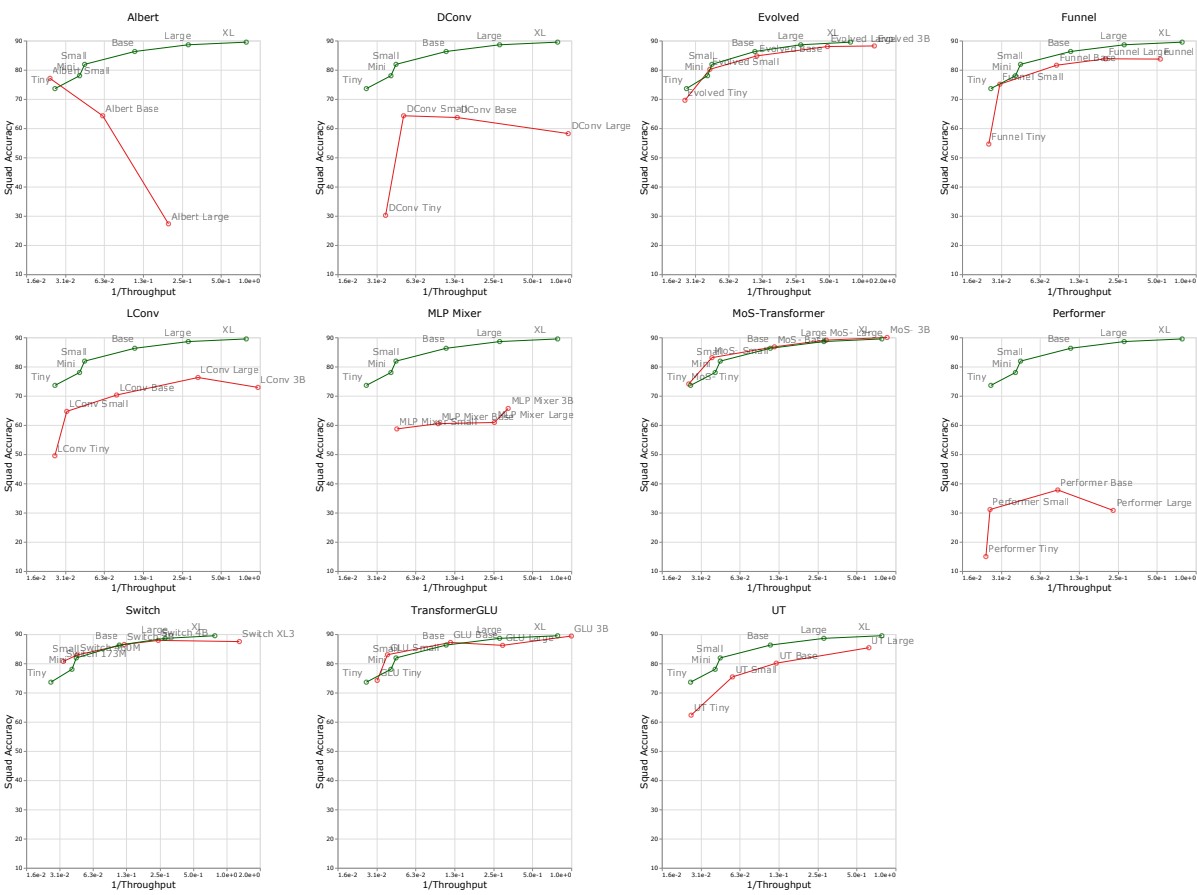

Figure 17: Quality-Throughput trade off for the downstream Squad Accuracy of vanilla Transformer compared to other models.