# OpenReview forum: "Scaling Laws vs Model Architectures: How does Inductive Bias Influence Scaling?"
_EMNLP/2023/Conference — EMNLP 2023 Findings_

### Official Review · Reviewer_1kqg · 2023-08-01

**Typos Grammar Style And Presentation Improvements:** 1. As discussed before, please make s…
**Soundness:** 3

**Excitement:**

3: Ambivalent: It has merits (e.g., it reports state-of-the-art results, the idea is nice), but there are key weaknesses (e.g., it describes incremental work), and it can significantly benefit from another round of revision. However, I won't object to accepting it if my co-reviewers champion it.

**Paper Topic And Main Contributions:**

This paper aims to find out the scaling properties on network architectures in transformer models. Specifically, they investigate the effect on upstream and downtream tasks, respectively. Experiments are done in different variants of transformers.

**Questions For The Authors:**

Can the authors provide the scaling effect only on the MLP layers? Beacuse the MLP is the fundamental architecture for neural networks.

**Reasons To Accept:**

1. The experiments are detailed and adequate.
2. Investigating the scaling behaviour of the neural networks is of importance to the community.

**Reasons To Reject:**

1. Although the experiments are adequate with detailed illustration, I cannot see the words or numbers clearly enough.
2. The metrics used in this paper is indirect with the learning dynamics of the transformers. I wonder how the loss is affected by the scaling behaviour, because the training process is direcly pushing the gradients w.r.t loss, instead of other metrics.
3. I suggest the authors clearly define all metrics used in the paper with mathematics formula.

**Reproducibility:**

3: Could reproduce the results with some difficulty. The settings of parameters are underspecified or subjectively determined; the training/evaluation data are not widely available.

**Reviewer Confidence:**

3: Pretty sure, but there's a chance I missed something. Although I have a good feel for this area in general, I did not carefully check the paper's details, e.g., the math, experimental design, or novelty.

---

> ### Author Rebuttal · Authors · 2023-08-29
>
> Thank you for taking the time to review our paper. We address your concerns below:
>
> Re: Vector graphics. All diagrams used in the paper are indeed vector graphics, and appear to be rendering correctly as such in Google Chrome.
>
> Re: Metrics. In Table 1 we do indeed show the negative log perplexity of the pretrained models, which is a function of the loss. Regardless, it has been known in the NLP community that pretraining perplexity does not necessarily correlate with downstream performance, as we additionally reaffirm in Table 2. Finetuning loss especially does not correlate with performance, as it could mean overfitting. In general, we base our evaluations off of downstream metrics as those are the metrics most important to practitioners.
>
> Re: Metric formulas. We only use standard common metrics and tasks in our work, and for particular datasets we refer to the original dataset papers in Section 4, that provide detailed descriptions of their metrics and averaging conventions.

---

### Official Review · Reviewer_MfUv · 2023-08-02

**Soundness:** 4

**Excitement:**

4: Strong: This paper deepens the understanding of some phenomenon or lowers the barriers to an existing research direction.

**Paper Topic And Main Contributions:**

This paper studies how the performance of different language model architectures changes with their scale (i.e. FLOPS or number of parameters) on both upstream and downstream tasks. The authors considered up to 10 different architectures and empirically showed that the latter can display very different scaling behaviors, thereby highlighting a new dimension to consider (i.e. model scale) when constructing/proposing novel architectures. In particular they showed that, although a given architecture might perform well at a given scale, its scaling behavior might be suboptimal.

The authors also demonstrated that the same architecture can display different scaling wrt. the upstream and downstream tasks; a result which mirrors previous findings, for fixed-size architectures.

**Questions For The Authors:**

Is there available some repository with your code somewhere?

**Reasons To Accept:**

- The paper highlights another dimension to consider when introducing new architectures for language modelling, namely their scale. Since the same architecture can perform differently on different scales, researchers who propose new architectures should consider reporting how their proposal scales with FLOPS (or number of parameters); as to characterize their proposal more thoroughly.

- The paper additionally provides a large set of results that can be used for comparison (i.e. as baselines). Specially by those researches who want to introduce novel architectures or mechanisms for language modelling (see point above).

I believe these two reasons are enough to make this paper an important contribution to EMNLP.

**Reasons To Reject:**

- The paper considers models up to 40B parameters. Yet the so-called *emergent properties* of language models seem to appear for larger model sizes (Wei et al. 2022). It’s therefore a valid question whether the scaling of different architectures breaks down after crossing the (architecture-dependent) critical scale of emergent properties. Indeed, it would have been interesting to additionally study this architecture-dependent critical scale itself.

- There is little to no details available to reproduce the results of the paper.

*Links to references*

Wei et al. (2022): https://arxiv.org/pdf/2206.07682.pdf


**Reproducibility:**

2: Would be hard pressed to reproduce the results. The contribution depends on data that are simply not available outside the author's institution or consortium; not enough details are provided.

**Reviewer Confidence:**

3: Pretty sure, but there's a chance I missed something. Although I have a good feel for this area in general, I did not carefully check the paper's details, e.g., the math, experimental design, or novelty.

---

> ### Author Rebuttal · Authors · 2023-08-29
>
> Thank you for your time in reading our paper, providing your valuable feedback, and for recognizing that this would be an important contribution to EMNLP!
>
> Re: 40B and emergent abilities. Emergent abilities as described by Wei et al. 2022 pertains only to few-shot tasks such as BIG-Bench Hard (BBH). Our work only studies the relationship between model scale, architecture and downstream finetuning performance. There are a variety of reasons that we formulate our study in this way as explained in further detail in our rebuttal to Reviewer rySs. In short, it allows us to consider a wider diversity of architectures than that of those known to exhibit in-context learning ability or emergent abilities, and we are able to compare them in a more controlled way. We would further add that for traditional finetuning performance, emergent abilities based on model size are not known to exist as evidenced by the SuperGLUE leaderboard which shows a more gradual relationship between increasing model size and finetuning performance, e.g. T5 11B and PaLM 540B is only 1pt apart.

---

### Official Review · Reviewer_rySs · 2023-08-08

**Soundness:** 3

**Excitement:**

3: Ambivalent: It has merits (e.g., it reports state-of-the-art results, the idea is nice), but there are key weaknesses (e.g., it describes incremental work), and it can significantly benefit from another round of revision. However, I won't object to accepting it if my co-reviewers champion it.

**Missing References:**

- Hoffmann et al. "Training Compute-Optimal Large Language Models", NeurIPS 2022

**Paper Topic And Main Contributions:**

This paper provides an extensive empirical evaluation of the scaling properties of different encoder-decoder models for seq2seq tasks including transformers, efficient variants thereof like Performer and ALBERT, and non-transformer architectures like Dynamic Convolutions and MLP-Mixer. In particular, the question under investigation is how the inductive bias due to the architecture affects the scaling of the perplexity loss as a function of model size (measured as flops, as the training dataset is fixed). The main finding of this effort (which spans a remarkable 100 different models ranging from 15M to 40B parameters) is that architecture indeed has an effect on the performance of the pretraining seq2seq task, and an even starker effect on downstream fine-tuning tasks (NLU tasks coming from GLUE and SuperGLUE and SQuAD). In addition, regular transformers are however mostly still the winner in regard of how performance scales as a function of model size.

### Changes after rebuttal
* soundness increased to good
* confidence increased to quite sure

**Questions For The Authors:**

- The vanilla transformer seems to be winning across the board, particularly in the downstream evaluations. On the other hand, that is also the architecture that has been around for longer. Would it be reasonable to hypothesize that this gives an advantage to transformers in terms of the training workflows (including optimizers, schedulers, normalization blocks, starting hyperparameter values, etc and their interplay) having grown adapted to this architectures for longer? In other words, were the community to spend as much time as it has training transformers on hyperoptimizing, say, MLP-Mixer, couldn't that change the reported results?
- It is a bit puzzling that the paper didn't mention anything about the effect of dataset size, and data-optimal scaling like the "chinchilla scaling", since data availability is starting to be the bottleneck in training LLMs, at least is some conditions, and the interplay between model size and dataset size is currently arguably the de facto criterion to allocate training budget. Is it still possible to convincingly justify the choice of ignoring the effect of scaling dataset size in this type of studies? Wouldn't it be reasonable to expect that different model architecture will also display different scaling depending on where they are located in terms of allocating flops to parameters rather than data? If that is the case, the main results of this study might change considerably at different data regimes.

**Reasons To Accept:**

- Very extensive empirical evaluation of the scaling properties across different model architectures spanning a range of 15M to 40B parameters
- The paper is clearly written and straightforward
- The paper presents a simple, yet solid and unified framework for the evaluation of a large variety of models

**Reasons To Reject:**

- The paper doesn't analyze a crucial factor determining the scaling properties of model performance which is dataset size, and doesn't even refer to the literature pertaining to data-optimal scaling laws like the so-called "Chichilla scaling"
- The paper is rich in empirical results, but unfortunately limited in terms of mechanistic explanations and actionable recommendation. With the exception of some passing observations like noticing the importance of cross-attention in the encoder to guarantee downstream performance, the paper does not provide many insights on what architectural modules support favorable scaling
- The paper stands out for its extensive empirical evaluation, but that is not accompanied by a detailed enough report of the computation and hardware budget that required this study. The paper mentions the use of TPU-v3 pods, but doesn't mention how many pods for how long had to be used in the study, and what the resulting energy and CO2-emission footprint was. This might be a concern for this type of large-scale studies. To be clear, this type of studies are clearly important also because, despite their possibly large footprint, clarifying models scaling could potentially help amortize that cost over the savings of future more efficient architectures. But in the current conjuncture of GPU scarcity, concerns over energy costs and growing sensitivity for climate issues, there seems to be a growing expectation that these points be explicitly addressed

**Reproducibility:**

2: Would be hard pressed to reproduce the results. The contribution depends on data that are simply not available outside the author's institution or consortium; not enough details are provided.

**Reviewer Confidence:**

4: Quite sure. I tried to check the important points carefully. It's unlikely, though conceivable, that I missed something that should affect my ratings.

---

> ### Author Rebuttal · Authors · 2023-08-29
>
> Thank you for your thorough review of our paper! These are great questions. We address your comments below:
>
> Re: Chinchilla optimality / Dataset size. The importance of pretraining dataset size is limited in this study, as unlike the setting studied in Hoffman et al. and many recent LLM works, we study the relationship between architecture, model size, and *downstream finetuning performance*, whereas Hoffman et al. studies language modeling / few-shot learning. For few-shot, evaluation is done directly on the pretraining checkpoint without finetuning, and thus pretraining convergence and the diversity of knowledge available in pretraining dataset matters. In our experiments, the finetuning tasks supply all models with exactly the same examples required to learn the task, and we are able to finetune each model until convergence to accommodate for different convergence behaviors.
>
> As such, we intentionally fix the pretraining experimental settings between all models for fair comparison and to narrow the search space: all models see the same number of pretraining tokens before finetuning. The pretraining setup for our encoder-decoders in this paper closely follows the well established ablation setting in Raffel et al. 2019. Since we follow the T5 setting, we also pretrain with span corruption, which has been shown to not perform well on few-shot learning (Tay et al. 2022, UL2), making Hoffman et al. especially not applicable.
>
> One benefit of comparing models in this way is precisely because it is less dependent on pretraining design to acquire in-context learning (ICL) ability. It is still an open question whether some of the architectures we study in this work can even exhibit ICL abilities in the same way as the normal Transformer, which seems like a separate open research question. For example, it is not possible to apply a token-level causal mask to a Funnel Transformer, making the usage of pure causal language modeling (CLM) pretraining objective not possible. This complicates comparisons with models that can indeed train purely with CLM, which works better for ICL. Designing the experiments as we did levels the playing field in this regard and allows us to focus solely on the comparison of inductive biases and scale.
>
> This is not to say that Scaling Laws vs. Model Architectures for ICL is not important, but it is simply outside the scope of this work and the diversity of architectures we evaluate. We are happy to change the framing and/or add this discussion to the paper to clarify in the final version, as well as reference Hoffman et al. An aside on Chinchilla optimality: the process proposed by Hoffman et al. requires plotting multiple isoFLOP curves at smaller FLOP bands in order to extrapolate to larger FLOP bands. Hoffman et al. only showed this extrapolation process to be accurate for their relatively vanilla Transformer architecture. In order for us to properly apply Hoffman et al. to our work here, we would have to not just derive compute-optimal scaling laws as they do for every architecture, but also empirically verify that the extrapolation assumptions of Hoffman et al. still hold for all architectures considered. This would also be a different research question than the one studied by this paper.
>
> Re: Lack of mechanistic explanations or actionable recommendations. We did indeed aim to provide an empirical study of parameter scaling multiple diverse architectures, and not a mechanistic interpretation of any particular architecture. We do believe though that there are many significant findings of interest to the community of which we thoroughly describe in the results and conclusions of our paper. Mainly, different architectures scale differently, and what might be the best architecture at one compute region may not be the best at another compute region. This has major implications for how researchers and practitioners should go about conducting and interpreting architecture research, especially as many architecture papers only show a narrow slice of the compute region.
>
> Re: Compute required for this study. Line 290 clearly states “We pretrain and finetune our models with 16 TPU-v3 chips with data parallelism,” Table 1 lists the training speed in steps per second for each model, and Line 278 states that we pretrain for 2^19 (524k) steps, which is sufficient to estimate the degree of compute required for this study.
>
> Re: Vanilla Transformer and hyperparameters. This is a true limitation of this study that we discuss in the limitations section. As finding the optimal way of training specific architectures is a never ending quest for the research community, it will never be possible to fully disentangle this. However, we have tuned each model to the best of our ability, reaching out to authors / experts when possible.
>
> Thanks again for your feedback!

---

### Official Review · Reviewer_n3js · 2023-08-09

**Typos Grammar Style And Presentation Improvements:** 1. In Figure 1, there are some circle…
**Soundness:** 4

**Excitement:**

4: Strong: This paper deepens the understanding of some phenomenon or lowers the barriers to an existing research direction.

**Missing References:**

None.

**Paper Topic And Main Contributions:**

The topic of this paper is the relationship between scaling laws and model architecture The authors attempt to explore whether inductive bias inside the model architecture affects the scaling performance.
The contributions of this paper are:
1. Comprehensive experiments on over 100 models in pretraining and finetuning to derive scaling laws for different model architectures.
2. They find that models perform well in one scale region and do not consistently be the best in another.  While the vanilla Transformer performs best in scaling.
3. They observe that some architectures are hard to scale up, such as the performer, suggesting that the linear Transformer may have a bottleneck in scaling.

Overall a good article with very detailed experiments and very valuable findings learned with a lot of computing budget, especially in the trend of LLMs. These findings will be very helpful for model selection in both research and engineering.

**Questions For The Authors:**

Question A: From line 387 to line 389, "A high αU,D value means that the transfer to the downstream tasks is better as a model scales." Isn't it more likely that upstream perplexity is not closely related to downstream metrics, rather than the model's transfer ability?

Question B: In Figure 4(b), there is a huge fluctuation in the performance of the MoS model from 16 to 36 layers, have you tried deeper layers like 100 to compare the results?

**Reasons To Accept:**

1. A huge number of experiments. Pretrain and finetune over 100 models (10 model architectures with different sizes) and get valuable findings in the scaling laws with different architectures.
2. Well-written paper.

**Reasons To Reject:**

1. The c4 dataset is the only one for all experiments and not sure whether the pretrain dataset will have an effect.
2. All the models are Encoder-Decoder arch while Decoder-only missed. But I believe this should not detract from the fact that this is a good paper that would contribute to the field.

**Reproducibility:**

2: Would be hard pressed to reproduce the results. The contribution depends on data that are simply not available outside the author's institution or consortium; not enough details are provided.

**Reviewer Confidence:**

3: Pretty sure, but there's a chance I missed something. Although I have a good feel for this area in general, I did not carefully check the paper's details, e.g., the math, experimental design, or novelty.

---

> ### Author Rebuttal · Authors · 2023-08-29
>
> Thank you for taking the time to review our paper and give your thoughtful comments!
>
> Re: Only C4 pretraining. As the main goal of the work is to derive a relationship between various model architectures and their scaling behavior, we intentionally decided to fix the pretraining experimental settings between all models for fair comparison. As C4 is a widely used and publicly available dataset, as well as one where many of our considered architectures developed their models on, it was a natural choice for this. While deriving the relationship between pretraining dataset, architecture, and scale would be very interesting, the cross product becomes infeasible, given the huge number of experiments we run as the reviewer mentioned.
>
> Re: Encoder-decoder vs decoder-only. Similar to the pretraining dataset, we decide to also fix this variable as many of the architecture variants we consider can be expressed as either encoder-decoder or decoder only. Some however, such as Funnel Transformer or MLP-Mixer are only applicable to the encoder side. This made encoder-decoder be a more suitable backbone for more of the modifications we wanted to evaluate.
>
> Question A: Depends on what the reviewer means by “transfer ability” but we see from Table 2 that the relationship between upstream and downstream performance does fluctuate quite a bit between architectures, showing that upstream performance does not necessarily transfer to strong finetuning performance.
>
> Question B: Unfortunately we did not, in order to bound the number of experiments, we tested all models only to the FLOP region shown in the paper.
>
> Finally, thank you for your notes on the presentation, we will revise the paper to improve it for the final version.
>
> Thanks again for your feedback!

---

### Meta-Review · Area_Chair_MRgN · 2023-09-14

**Recommendation:** 3

**Metareview:**

Pros:
- Extensive experiments with many models.
- No reviewers mentioned this, but I think that the discussion of how hyperparameter settings can change scaling laws could be very valuable on a practical level, as it could support hyperparameter selection decisions.

Cons:
- They only consider encoder-decoder models. Because the architectures are limited, there is a missed opportunity to consider inductive biases of radically different architectural approaches.
- The analysis and discussion is limited. Given that the paper is framed as an exploration of inductive bias, there is little to no discussion of the source or nature of those inductive biases when using different architectures. This makes the title a little misleading; perhaps it should be called “How Does Architecture Influence Scaling” instead.
- They only consider up to 40B parameter models, which is below the range claimed to show dramatic breakthroughs on capabilities like chain of thought.

---

### Decision · Program_Chairs · 2023-10-07

**Decision:**

Accept-Findings

**Comment:**

Pros:
- Extensive experiments with many models.
- No reviewers mentioned this, but I think that the discussion of how hyperparameter settings can change scaling laws could be very valuable on a practical level, as it could support hyperparameter selection decisions.

Cons:
- They only consider encoder-decoder models. Because the architectures are limited, there is a missed opportunity to consider inductive biases of radically different architectural approaches.
- The analysis and discussion is limited. Given that the paper is framed as an exploration of inductive bias, there is little to no discussion of the source or nature of those inductive biases when using different architectures. This makes the title a little misleading; perhaps it should be called “How Does Architecture Influence Scaling” instead.
- They only consider up to 40B parameter models, which is below the range claimed to show dramatic breakthroughs on capabilities like chain of thought.